# High-Resolution Linear Epitope Mapping of the Receptor Binding Domain of SARS-CoV-2 Spike Protein in COVID-19 mRNA Vaccine Recipients

Yuko Nitahara,[a] Yu Nakagama,[a] Natsuko Kaku,[a] Katherine Candray,[a] Yu Michimuko,[a] Evariste Tshibangu-Kabamba,[a] Akira Kaneko,[a] Hiromasa Yamamoto,[b] Yasumitsu Mizobata,[b] Hiroshi Kakeya,[c] Mayo Yasugi,[d,e,f] Yasutoshi Kido[a]

aDepartment of Parasitology & Research Center for Infectious Disease Sciences, Graduate School of Medicine, Osaka City University, Osaka, Japan

bDepartment of Traumatology and Critical Care Medicine, Graduate School of Medicine, Osaka City University, Osaka, Japan

cDepartment of Infection Control Science, Graduate School of Medicine, Osaka City University, Osaka, Japan

dDepartment of Veterinary Science, Graduate School of Life and Environmental Sciences, Osaka Prefecture University, Izumisano, Osaka, Japan

eOsaka International Research Center for Infectious Diseases, Osaka Prefecture University, Izumisano, Osaka, Japan

fAsian Health Science Research Institute, Osaka Prefecture University, Izumisano, Osaka, Japan

**ABSTRACT**   The prompt rollout of the severe acute respiratory syndrome coronavirus 2 (SARS-CoV-2) mRNA vaccine is facilitating population immunity, which is becoming more dominant than natural infection-mediated immunity. In the midst of coronavirus disease 2019 (COVID-19) vaccine deployment, understanding the epitope profiles of vaccine-elicited antibodies will be the first step in assessing the functionality of vaccine-induced immunity. In this study, the high-resolution linear epitope profiles of Pfizer-BioNTech COVID-19 mRNA vaccine recipients and COVID-19 patients were delineated by using microarrays mapped with overlapping peptides of the receptor binding domain (RBD) of the SARS-CoV-2 spike protein. The vaccine-induced antibodies targeting the RBD had a broader distribution across the RBD than that induced by the natural infection. Half-maximal neutralization titers were measured *in vitro* by live virus neutralization assays. As a result, relatively lower neutralizability was observed in vaccine recipient sera, when normalized to a total anti-RBD IgG titer. However, mutation panel assays targeting the SARS-CoV-2 variants of concern have shown that the vaccine-induced epitope variety, rich in breadth, may grant resistance against future viral evolutionary escapes, serving as an advantage of vaccine-induced immunity.

**IMPORTANCE** Establishing vaccine-based population immunity has been the key factor in attaining herd protection. Thanks to expedited worldwide research efforts, the potency of mRNA vaccines against the coronavirus disease 2019 (COVID-19) is now incontestable. The next debate is regarding the coverage of SARS-CoV-2 variants. In the midst of vaccine deployment, it is of importance to describe the similarities and differences between the immune responses of COVID-19 vaccine recipients and naturally infected individuals. In this study, we demonstrated that the antibody profiles of vaccine recipients are richer in variety, targeting a key protein of the invading virus, than those of naturally infected individuals. Vaccine-elicited antibodies included more nonneutralizing antibodies than infection-elicited antibodies, and their breadth in antibody variations suggested possible resilience against future SARS-CoV-2 variants. The antibody profile achieved by vaccinations in naive individuals provides important insight into the first step toward vaccine-based population immunity.

**KEYWORDS** SARS-CoV-2, spike, neutralizing antibodies, serology, COVID-19, RBD, immunoserology, spike protein

Address correspondence to Yasutoshi Kido, kido.yasutoshi@med.osaka-cu.ac.jp.

**G**lobally, mRNA vaccines have prevailed to mitigate the coronavirus disease 2019 (COVID-19) pandemic. Given the prompt progress in the development of vaccines and their fast rollout at a global scale, population immunity against the severe acute respiratory syndrome coronavirus 2 (SARS-CoV-2) will largely depend on vaccine-induced rather than the infection-induced immunity. In this start of acquiring vaccine immunity as a society against COVID-19, the *de novo* repertoire of vaccine-elicited antibodies in SARS-CoV-2 infection-naive individuals will be the first step to build an optimal host defense system toward vaccine-based population immunity.

Currently, the efficacy of vaccine-induced immunity against SARS-CoV-2 in an individual is evaluated by potential surrogate markers, such as half-maximal neutralization titers (NT50s) using live or pseudotyped viruses and total antibodies titers against the receptor binding domain (RBD) of the spike protein of the virus (1–4). Understanding the epitope profile of both vaccine recipients and naturally infected individuals can readily help elucidate the molecular basis of these markers as a surrogate. Moreover, the coevolution of vaccine-induced host immunity and virus escape will be one of the most important elements to consider in the way of achieving herd immunity against COVID-19.

The RBD of the spike protein of SARS-CoV-2 is widely considered the key protein target for designing vaccines and developing neutralizing antibodies as therapeutic agents (5, 6). Epitope profiles of sera from individuals naturally infected with COVID-19 have enabled the identification of several immunodominant regions in the spike protein (7–9). While most immunodominant epitopes are located outside the RBD, the minor proportion targeting specifically the neutralizing RBD epitopes explain the majority of viral neutralizability and protection against reexposures (10, 11). In fact, neutralizing monoclonal antibodies (NAbs) developed as potential therapeutics also target mainly the epitopes located in the RBD (6, 10, 12–15). While a growing number of individuals acquire vaccine immunity, the detailed epitope profile of the humoral immune response to the mRNA vaccine is not fully understood (1, 16, 17).

In this study, high-resolution linear epitope profiling targeting the RBD was performed using sera of both mRNA vaccine recipients and COVID-19 patients. By comparing the epitope profiles, we sought to describe the similarities and differences between the humoral immune responses induced by BNT162b2 mRNA (Pfizer/BioNTech) vaccination and natural infection. Information provided by this study will be crucial in this postvaccine era of the COVID-19 pandemic.

## RESULTS

**Total IgG titers targeting the RBD and neutralization assay using live SARS-CoV-2.** All vaccine recipients ($n = 21$) and COVID-19 patients ($n = 20$) revealed seropositivity to anti-RBD IgG according to the manufacturer's threshold (>50 arbitrary units [AU]/mL), and the 2 groups did not show significant differences in their levels of anti-RBD IgG titers (Fig. 1a). However, the neutralization assay using live SARS-CoV-2 showed remarkably lower NT50s in vaccine recipients than in COVID-19 patients ($P = 0.0035$) (Fig. 1b). The ratio between the anti-RBD IgG antibody titer and the NT50 value was calculated in individuals as shown in Fig. 1c. It appeared that the anti-RBD IgG/NT50 ratios were significantly higher in vaccine recipients than those in COVID-19 patients ($P < 0.001$) (see Fig. 3c). This result indicated that the sera of vaccine recipients were more abundant in nonneutralizing, mere binding IgG antibodies, suggesting a discrepancy in the epitope profiles between vaccine recipients and COVID-19 patients. None of the vaccine recipients were seropositive to anti-N IgG, ensuring that they were naive to SARS-CoV-2 infection (see Table S1 in the supplemental material).

**Comparison of linear epitope profiles targeting the RBD of vaccine-elicited and infection-elicited sera.** To delineate the discrepancy in the epitope profiles between vaccine recipients and COVID-19 patients with high resolution, we next mapped and compared the immunodominant epitopes of representative sera chosen from both groups by using an overlapping 15-mer linear-peptide array (Fig. 2a). The representative sera were selected from vaccine recipients ($n = 5$; median age, 37 years) and

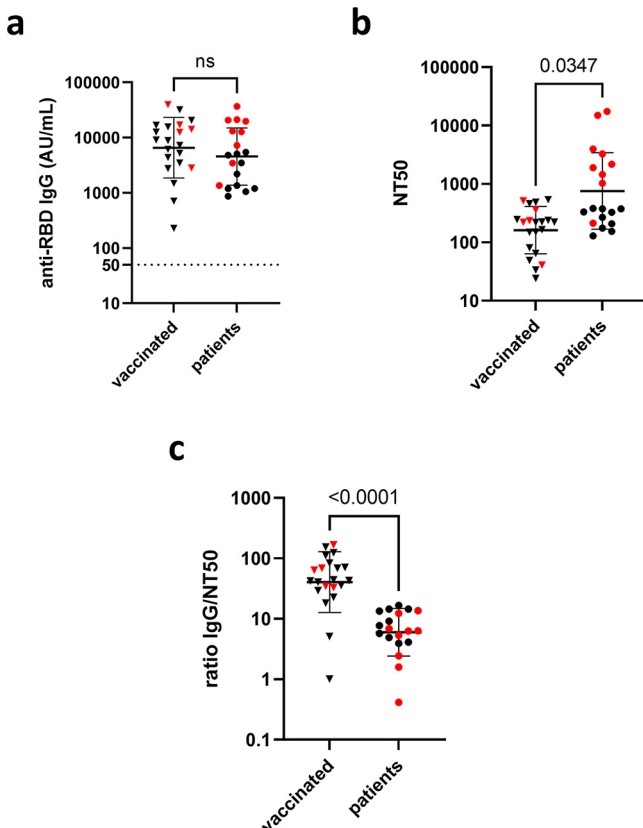

**FIG 1** Total antibody titers targeting the RBD and neutralization of live SARS-CoV-2. (a) Anti-RBD IgG titers of vaccine (BNT162b2) recipients ($n$ = 21) and COVID-19 patients ($n$ = 19) were depicted. No significant difference in the level of total anti-RBD IgG titers was observed. (b) The half-maximal neutralization titers (NT50s) were remarkably lower in the vaccinated group than those in patients. (c) The anti-RBD IgG/NT50 ratio was plotted for both groups. Black horizontal bars indicate geometric mean with geometric standard deviation. Red dots denote the subset of samples chosen for the microarray analysis. For detailed information on subjects, see Table S1.

COVID-19 patients ($n$ = 10; median age, 58 years) based on their anti-RBD antibody titers so that the distribution of titers would represent the original population ($P$ = 0.27 and $P$ = 0.12, respectively) (Table S1). Selected specimens are depicted as red dots in Fig. 1. The sera were incubated with the designed microarrays arranged with 15-mer overlapping peptides of the RBD on the surface (Fig. 2a). The designated array did not show any considerable unspecific binding of the secondary antibody.

We generated a heatmap according to the relative signals of the overlapping peptides (Fig. 2b). When calculating the relative signals, we set an upper limit of 20,000 for raw signals obtained in the assay for a better visualization in the heatmap. Four out of 1,065 peptides in total exceeded the limit (see Table S3 in the supplemental material). Also, Z-scores of each peptide were compared individually (Fig. S1 in the supplemental material) and per group (Fig. 3a and b). Comparing the epitope profiles of the two groups, we identified two types of epitopes, as follows: (i) epitopes recognized by both groups and (ii) epitopes recognized only by vaccine-elicited sera. We identified epitopes based on the following criteria: to have visibly detectable peaks compared with adjacent peptides in the Z-score depiction (Fig. 3c) and to include oligonucleotide peptides reported previously as epitopes of NAbs in the literature (18, 19).

Overall, seven linear epitopes were recognized within the RBD, including four that were recognized by both groups (N394 to A411, peptide no. 26 to 27; T415 to F429, peptide no. 33; V433 to N450, peptide no. 39 to 40; R457 to S477, peptide no. 47 to 49) and three that were recognized by vaccine-elicited sera (N334 to A348,

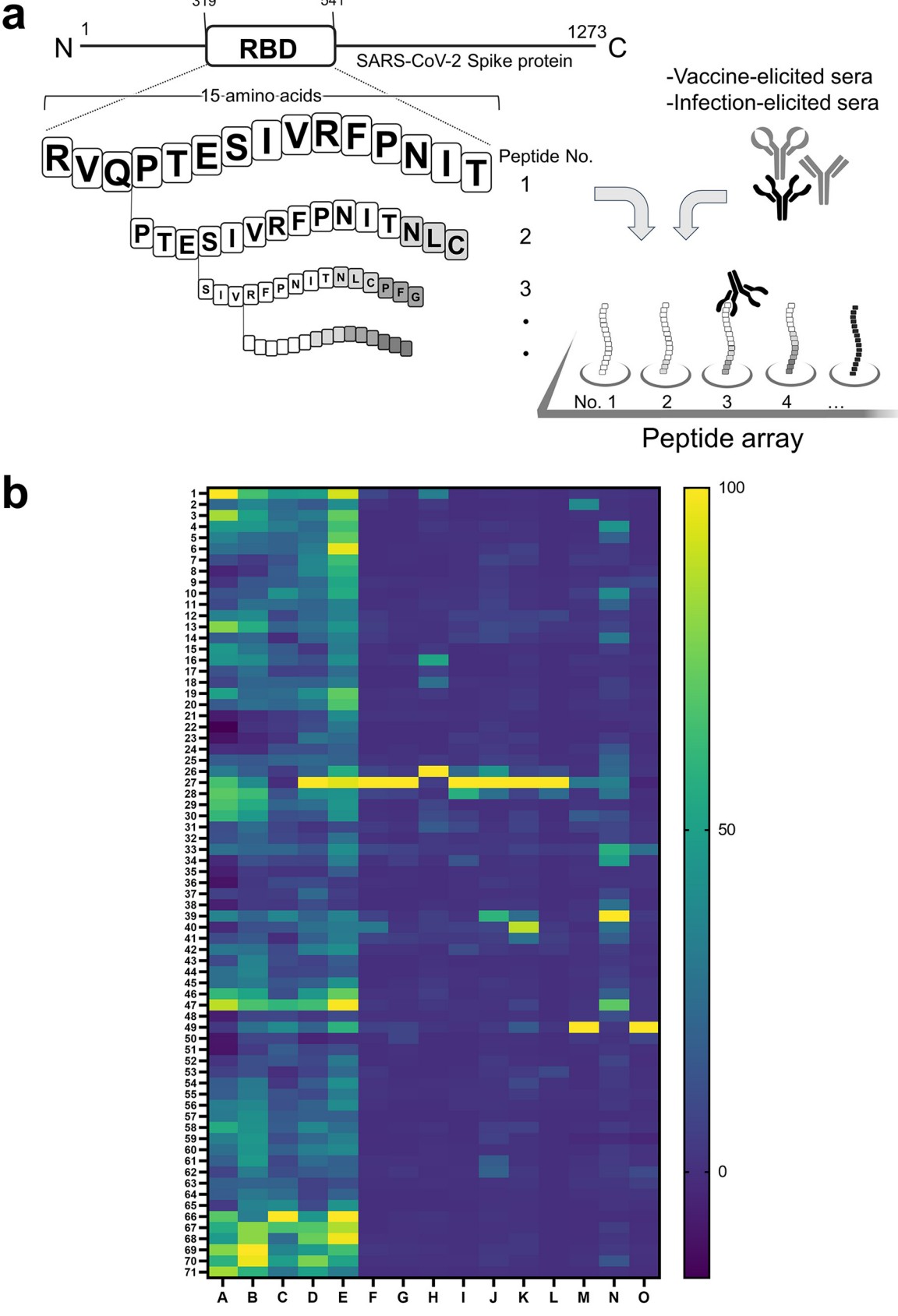

**FIG 2** High-resolution linear epitope mapping of the receptor binding domain (RBD) of the SARS-CoV-2 spike protein. (a) Overlapping 15-mer peptides (shift by 3 amino acids) of the RBD was sequentially synthesized on a cellulose membrane. Sera of vaccine recipients

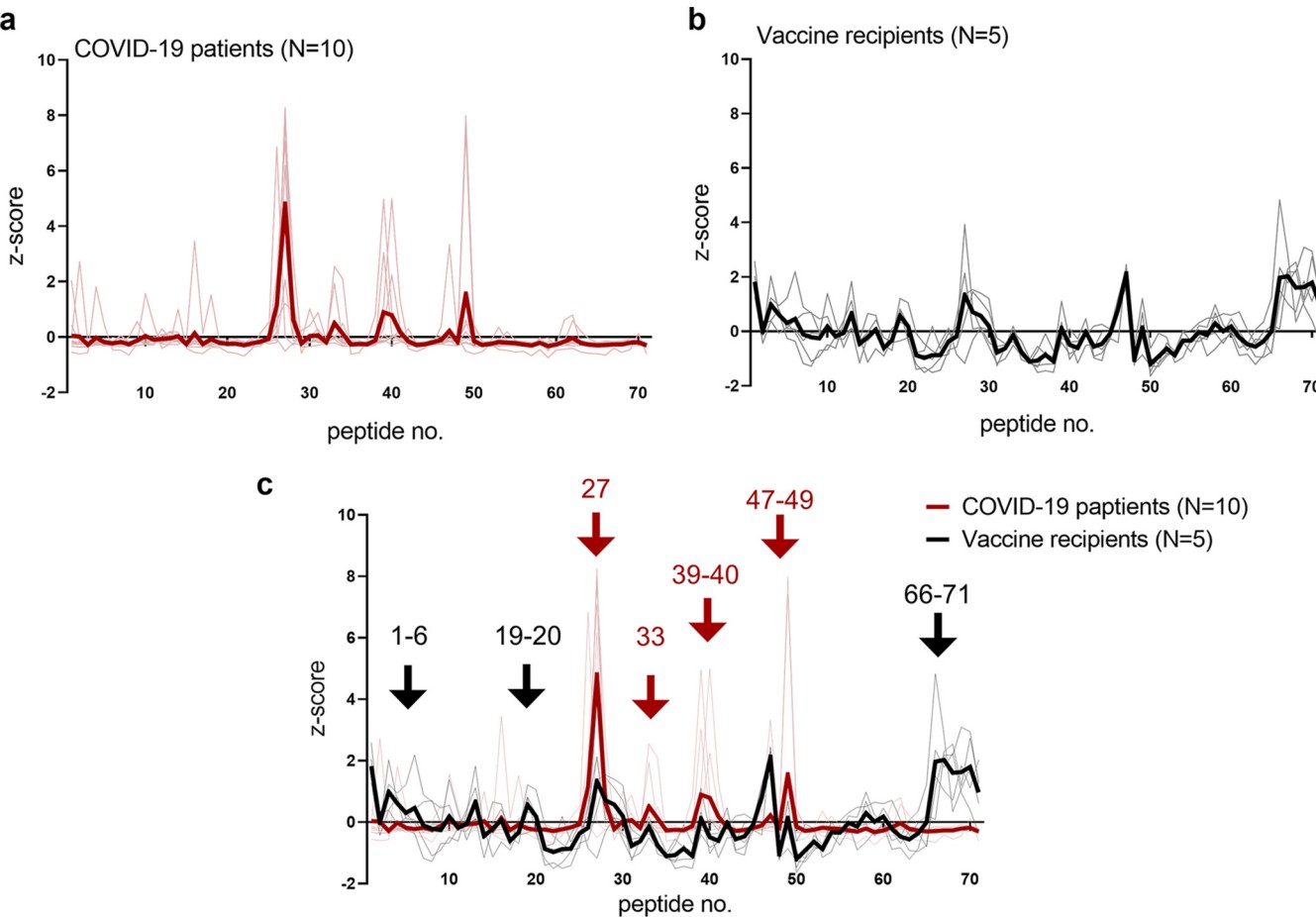

**FIG 3** Comparison of epitope profiles between the following two groups: BNT162b2 vaccine recipients (*n* = 5) and COVID-19 patients (*n* = 10). (a) Thin red lines denote peptide signals of individuals. Bold red lines depict the mean values of the peptide signals of the COVID-19 patient sera (*n* = 10). (b) Thin gray lines denote peptide signals of individuals. Bold black lines denote the mean values of the peptide signals of the vaccine recipient sera (*n* = 5). (c) Red arrows denote epitopes recognized in the sera of both groups. Black arrows denote epitopes identified only in the vaccine recipient sera. Designated peptide numbers are shown above the arrows.

peptide no. 6; S373 to L390, peptide no. 19 and 20; S514 to F541, peptide no. 66 to 71) (Fig. 2b and 3c).

**Epitopes recognized by both groups.** A total of four linear epitopes were recognized in both groups (Fig. 2b and 3c). Three (peptide no. 33, no. 39 to 40, and no. 47 to 49) of them shared the epitope regions of the RBD with neutralizing monoclonal antibodies reported previously as class 1 and class 3 (18).

Linear epitopes were identified at peptide no. 33 and peptide no. 47 to 49 (Fig. 3, Fig. 4a and b), sharing the epitopes with reported class 1 neutralizing antibodies (18). Also, peptide no. 39 to 40 (Fig. 3 and 4c) had epitope residues very similar to human monoclonal antibody REGN10987, categorized as a class 3 neutralizing antibody which sterically hinders the interaction between angiotensin converting enzyme 2 (ACE2) and the RBD (18, 20).

The linear epitope, peptide no. 26 to 27, was reactive at the highest level in most serum samples, as sera of 2/5 vaccine recipients and 7/10 patients had maximum reactivity to this peptide (Fig. 3c, Fig. S1). However, antibodies binding to this epitope seemed not

**FIG 2** Legend (Continued)
and COVID-19 patients were incubated with the microarray, followed by the procedure mentioned in the Materials and Methods to detect the reactive peptides. (b) Heat map identifying peptides recognized by IgG, IgA, and IgM in sera of vaccine recipients (samples A to E) and COVID-19 patients (samples F to O). The signal of each peptide was normalized to the maximum signal in each subject. Raw signal the data can be found in Table S3.

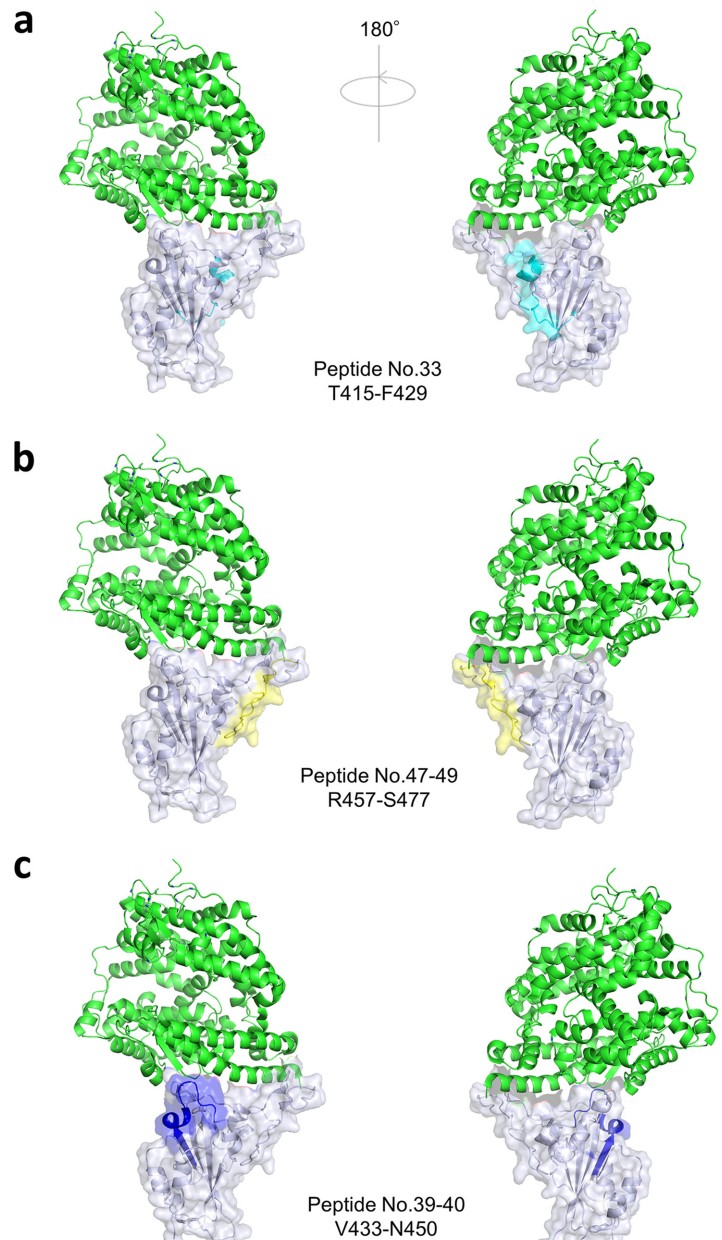

**FIG 4** Linear epitopes in the receptor binding domain (RBD) identified in both groups of vaccine recipients and COVID-19 patients. Angiotensin converting enzyme2 (ACE2), green; the RBD, tint blue; linear epitope T415 to F429 (peptide no. 33), cyan (a); linear epitope R457 to S477 (peptide no. 47 to 49), yellow (b); linear epitope V433 to N450 (peptide no. 39 to 40), blue (c).

to contribute to neutralizing the live virus, based on our observations detailed below. This peptide was found to be equally reactive, at a high extent, to a serum with negligible neutralizability, obtained from a COVID-19 patient who had undergone rituximab treatment (details found in Fig. S2 in the supplemental material) (21). In the RBD structure, the no. 27 peptide is located inside the core $\beta$ sheets, which is not exposed to the surface of the RBD in either an "up" or "down" position. Judging from the structural composition, this linear epitope would not affect ACE2 binding (Fig. S2).

**Epitopes recognized only in vaccine recipient sera.** Three linear epitopes of the RBD were uniquely found in vaccine recipient sera (Fig. 2b, Fig. 3b and c), and two of them (peptide no. 6 and no. 19 and 20) shared the epitope regions of the RBD with neutralizing monoclonal antibodies known as class 3 and class 4 (18).

Microbiology
Spectrum

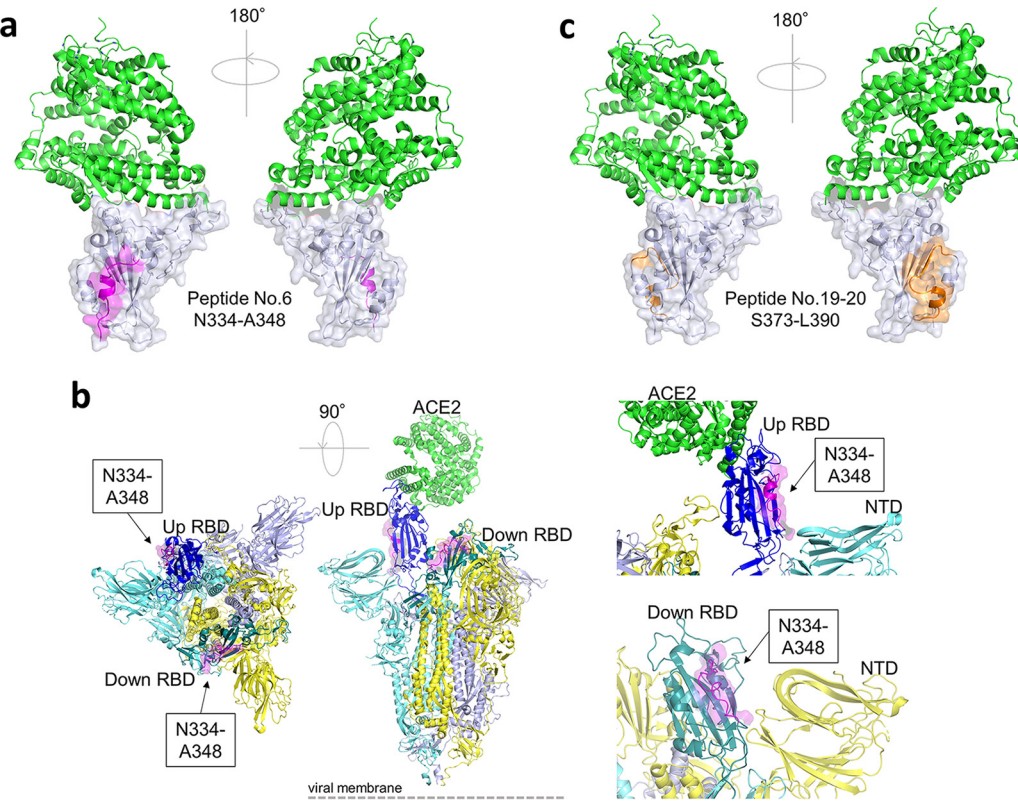

**FIG 5** Linear epitopes in the receptor binding domain (RBD) identified only in vaccine recipient sera. Angiotensin converting enzyme2 (ACE2) is shown in green. (a) The RBD is shown in tint blue. Linear epitope N334 to A348 (peptide no. 6), magenta. (b) SARS-CoV-2 spike trimer in one open, two closed (one RBD up, two RBD up) composition. Spike subunit 2 and N-terminal domain are in the same color, light blue, yellow, and tint blue. Up RBD is in dark blue. Down RBDs are in yellow. Linear epitope N334 to A348 has good accessibility in both the up and down composition of the RBD. (c) The RBD, green. Linear epitope S373 to L390 (peptide no. 19 and 20), orange.

At the N terminus of the RBD, namely, the peptide no. 1 to 6, we identified an epitope region detected only in the postvaccination sera (Fig. 2b and 3c). Especially, peptide no. 6 was an epitope of note, which could be recognized by the class 3 NAb S309 by P337 to A344 helix residues (Fig. 5a) (22). The epitope is distinct from the receptor-binding motif and has a good accessibility both in the up and down compositions of the RBD (PDB, 7A49) (Fig. 5b). Reactive peptide no. 13 was located close to the peptide no. 6 in the steric conformation (Fig. 2b and 3c). The peptide no. 13 contained six residues (K356 to C361), which share the conformational epitope of the NAb S309 with the peptide no. 6 (18, 22). Nevertheless, the peptide no. 13 alone does not seem to make a dominant epitope of S309.

Another identified epitope, peptide no. 19 and 20 (Fig. 5c), shared epitope residues with a neutralizing monoclonal antibody, CR3022, categorized as class 4, isolated from a SARS-CoV convalescent case (23, 24). This class 4 neutralizing antibody attaches to the RBD but distal to the ACE2 binding site and is highly conserved among different CoV species (25).

The third epitope, located at the peptide no. 66 to 71, did not match with any known monoclonal antibodies. Yi et al. detected the same region of the peptides (V524 to F541) as reactive from COVID-19 convalescent-phase serum in their linear epitope analysis (11). They also demonstrated that these peptides interacted with control sera as well (11). Thus, we considered our results on the corresponding peptides to be nonspecific.

**Linear epitopes mapping with single mutations found in SARS-CoV-2 variants.** Our analysis included single amino acid mutations of the RBD that are reported in the SARS-CoV-2 variants of concern, including B.1.1.7, B.1.351, and P.1 (PANGO lineage [26]).

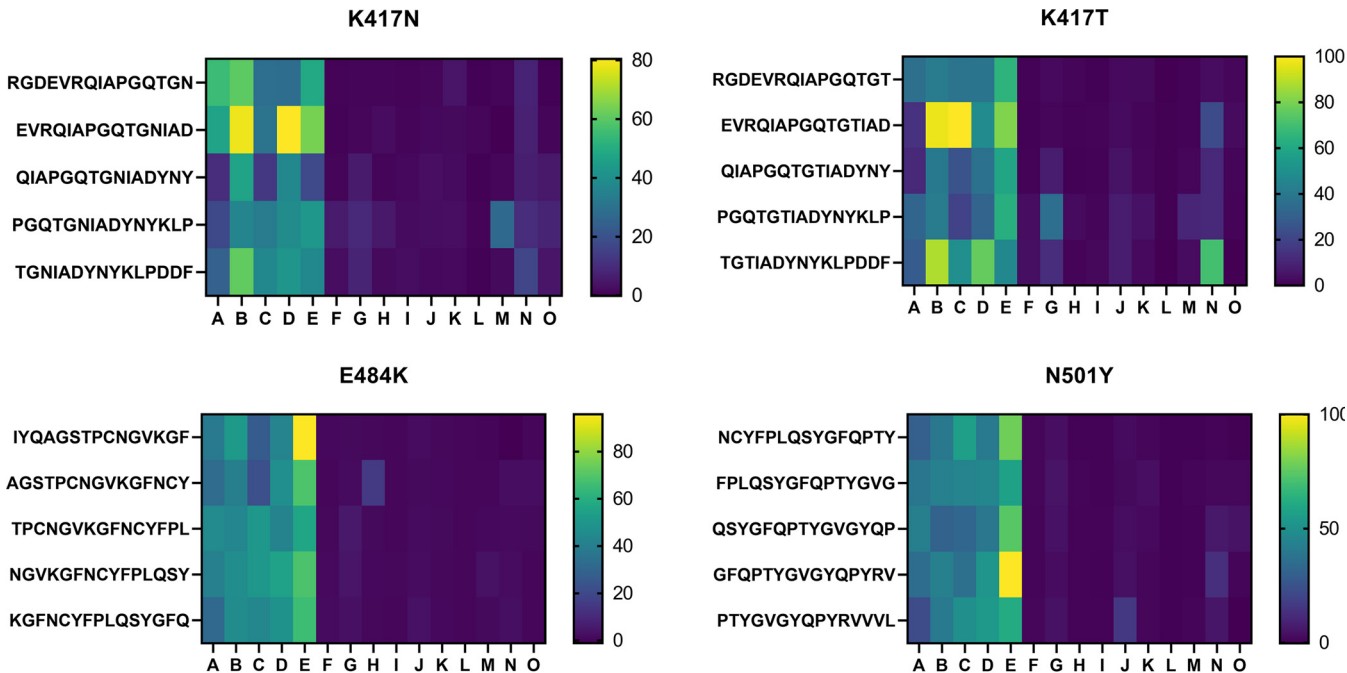

**FIG 6** Mutation peptide panels showed more reactivity to vaccine recipient sera than to patient sera. Heat maps identifying peptides carrying the single mutations, recognized by IgG, IgA, and IgM in sera of vaccine recipients (samples A to E) and COVID-19 patients (samples F to O). The signal of each peptide was calculated as a relative value to the maximum signal of each subject of 100.

Additional 15-mer peptides with the substituting amino acids K417N, K417T, E484K, and N501Y were incubated with both vaccine recipient and patient sera. Interestingly, vaccine-induced sera showed consistent signals to the mutated peptides, whereas patient sera had almost no reaction (Fig. 6).

In our original peptide array (Fig. 2b and 3), the overlapping peptides carrying the mentioned mutations sites (K417, E484, and N501 of the RBD) corresponded to peptide no. 29 to 33, no. 52 to 56, and no. 57 to 61, respectively. The peptides no. 52 to 61, containing E484 and N501, did not show any significant reaction in both vaccine recipients and COVID-19 patients (Fig. 2b and 3). The peptides no. 29 to 33 containing K417, which was considered an escape mutation, was reactive to antibodies in both patient and vaccine recipient sera as mentioned above (Fig. 2b and 3).

## DISCUSSION

This study revealed the linear epitope profiles targeting RBD elicited by BNT162b2 mRNA vaccination and natural infection of SARS-CoV-2. Our principal finding was that the variation of linear epitopes was broader in vaccine-elicited antibodies than that in infection-elicited antibodies, which may contribute to potent neutralization and thus resistance of the vaccine-elicited antibodies against the SARS-CoV-2 variants of concern.

Now, four categories of NAb classes are proposed to characterize the mode of recognition and epitope specificity (18). Class 1 NAbs block several proximal sites in the receptor binding motif (RBM) of the RBD and directly block ACE2 binding (18); class 2 NAbs recognize both up and down formations of the RBDs and epitope overlapping or close to ACE2-binding site (18); class 3 NAbs recognize both up and down RBD and bind outside the ACE2-binding site (18, 22); and class 4 NAbs bind only to up RBDs and do not directly block ACE2 binding, but destabilize the virus prefusion spike conformation (24, 25). Many of the human-isolated NAbs target RBD, while some target the N-terminal domain of the subunit 1 spike protein (14, 27). In our study, two classes of NAbs exclusively relevant in vaccine-elicited sera were found to be of specific note, namely, peptide no. 6 (Fig. 5a and b) targeted by the class 3 NAb (22) and peptide no. 19 to 20 (Fig. 5c) targeted by the class 4 NAb (24, 25). These epitopes locate outside

the ACE2-binding RBM (Fig. 5), while epitopes detected commonly in both vaccine- and infection-elicited repertoires clustered adjacent to the ACE2-binding site (Fig. 4).

The majority of NAbs targeting the RBM, which correspond to class 1 and 2, have been shown to exhibit decreased neutralization against the virus variants (15, 17, 28). For example, antibodies recognizing the linear epitope, here included in peptide no. 33, would possibly fail in neutralizing variants with the K417N mutation as described previously (15, 17, 28, 29). To the contrary, the linear peptides no. 6 and no. 19 and 20 (Fig. 5a, b, and c), corresponding to epitopes found exclusively in vaccine-elicited sera (Fig. 2b and 3c), revealed corresponding epitopes targeted by human NAbs isolated from SAR-CoV convalescent cases (S309 and CR3022, respectively) (22, 24, 25). These cross-neutralizing antibodies, belonging to class 3 and 4, recognize linear epitopes highly conserved among different CoV species. The epitopes recognized by these class 3 and class 4 NAbs are major contributors broadening the repertoire of vaccine-induced immunity. Located remotely from the RBM, such neutralizing epitopes stay rather free from variants (15, 17), and explain the resistance of vaccine-elicited sera toward viral mutational escapes (30, 31). The vaccine recipients' broader epitope profile spanning across the RBD may give immunological flexibility and resilience against this evolving virus. Our mutation peptide panels have also presented a rather optimistic view on discussing the efficacy of vaccine-induced immunity to efficiently recognize the SARS-CoV-2 variants (Fig. 6). However, considering that the linear epitope profiles harboring the mutation loci were not dominant in either vaccine sera or patient sera (Fig. 2b, Fig. 3b and c) and an abundance of conformational epitopes were found adjacent to the RBM, the extent to which these specific linear epitopes contribute in net neutralizability remains to be determined (10, 11, 14, 16).

We observed discrepancy between the neutralizability of sera obtained from vaccine recipients and patients, which could be partially explained by the difference in the time course of epitope selection and immune maturation. When comparing convalescent-phase sera and vaccine-elicited immunity, we found that the distribution of neutralizing epitopes was less generalized and focalized at specific peptides (Fig. 3; individual epitope distribution can be found in Fig. S1). Among the two modes of acquired immunity, our results indicate that infection-induced humoral immunity had established a more mature, finely selected antibody repertoire. Our snapshot observations are in line with the ideas that the maturation of infection-provoked repertoires occurs as early as 10 to 20 days after onset, or even earlier in the case of COVID-19 beginning at 4 to 7 days after onset (32, 33). The positive selection of relevant epitopes and the maturation of an antibody repertoire thus may lag behind in vaccine-induced immunity. Nevertheless, in this study, sera were sampled during the peak period of immune reaction in the host for both groups. Longitudinal evaluation of the epitope profiles and serological markers are needed for assessing host immune evolution and drawing conclusions to the above speculations.

In conclusion, we evaluated the similarity and difference in humoral immunity elicited by both the BNT162b2 mRNA vaccine and natural infection of SARS-CoV-2. High-resolution linear epitope profiles revealed the characteristic distribution of polyclonal antibodies spanning the RBD in vaccine recipient sera, which possibly accounted for the discrepancy observed in serological markers. Based on the multiplicity of neutralizing epitopes supporting the protectivity of vaccine-elicited antibodies, mRNA vaccine-elicited humoral immunity may harbor advantages in resisting the rapidly evolving pathogen.

There are several limitations in our study. The primary objective of our study was to compare the epitope profiles induced by vaccine and natural infection. However, the disease severity of the COVID-19 patients evaluated in the epitope analysis was skewed toward higher severity (one mild, two moderate, and seven critical patients), whereas the vaccine recipients were relatively healthy without major comorbidities. Indeed, the epitope profiles of critical patients (P01 to P07) seemed to differ from the ones of mild/moderate patients (P08 to P10) (Fig. 2b), suggesting the potential discrepancy in

COVID-19 immune response depending on patient clinical conditions. Their ages were equally distributed in both groups. This analysis was focused exclusively on the linear epitope profile targeting RBD. Moreover, the methodological limitation of the peptide binding assay using microarrays was that the results were semiquantitative. Therefore, as mentioned in the results, the reactive epitopes were characteristically selected by combining local peak detection and steric conformation in the structure with NAbs (34, 35). In this manner, unreported and nondominant epitopes could be overlooked in this study. Experimental observations on compositional epitopes and epitopes outside the RBD region were not made in this study. Nonetheless, our results reporting the mRNA vaccine's broad RBD epitope variety are in concordance with preceding reports (30, 36).

## MATERIALS AND METHODS

**Serum collection.** Two groups were analyzed in this study. One group included was vaccine recipients, who all received two doses of the BNT162b2 mRNA vaccine (Pfizer/BioNTech) with a 3-week interval ($n = 21$; age, 20 to 80 years old). Blood samples were obtained 17 to 28 days after the second dose. The second group included mild to critical COVID-19 patients; their infectivity status was confirmed by nucleic amplification testing ($n = 20$; age, 20 to 80 years old). We excluded asymptomatic patients in this study to observe the systematic immune response to SARS-CoV-2. (37) The blood collection of these patients was performed between 10 and 63 days (median 39 days) after the onset of disease. Detailed information of the subjects and severity of the disease of the patients can be found in Table S1 (38).

Blood samples were obtained by venipuncture in serum separator tubes, and the serum fraction was stored at $-80°C$. All subjects provided written consent before participating in this study. This study was approved by the institutional review board.

**Anti-RBD IgG quantification by chemiluminescent immunoassay.** Anti-RBD IgG titers of both groups were quantitated by measuring the chemiluminescence generated in the reaction mix containing serum IgG-bound, RBD-coated microparticles and acridinium-labeled anti-human IgG (Abbott SARS-CoV-2 IgG II Quant assay, USA) (39). Antibodies targeting the viral nucleocapsid protein (Anti-N IgG) were also measured for the sera of vaccine recipients to screen unrecognized exposure to SARS-CoV-2 (Abbott SARS-CoV-2 IgG assay) (40).

**Neutralization assay using live SARS-CoV-2.** The neutralization assay was carried out as described previously (41) but with modifications. Heat-inactivated (at 56°C for 45 minutes) vaccine recipient and patient sera and a SARS-CoV-2 negative-control serum were serially diluted 4-fold with Dulbecco's modified Eagle medium with 2% fecal bovine serum (2% FBS-DMEM) and incubated with 150 focus-forming units of SARS-CoV-2 JPN/TY/WK521 strain live virus particles (National Institute of Infectious Diseases, Japan) at 37°C for 1 hour. The monolayer of VeroE6 cells (National Institutes of Biomedical Innovation, Health and Nutrition) were then absorbed with the mixtures at 37°C. After a 1-hour incubation, the mixtures were replaced with fresh 2% FBS-DMEM. After an 8-hour incubation at 37°C, infection rates of the cells were determined by immunofluorescent staining, as follows. After fixation (4% paraformaldehyde, 15 minutes), cells were permeabilized (0.1% Triton X-100, 15 minutes) and incubated with rabbit anti-spike monoclonal antibodies (Sino Biological, China) (1:1,000, 1 hour at 37°C). Cells were then washed and incubated with Alexa488-conjugated goat anti-rabbit IgG (ThermoFisher Scientific, USA) (1:500, 45 minutes at 37°C). Antigen-positive cells were counted under a fluorescence microscope, and the percentage of neutralization was estimated as the viral infectivity under serum-treated conditions compared with that without serum.

**Epitope mapping of the RBD.** For precise linear epitope screening, overlapping 15-mer peptides (shift by 3 amino acids) were sequentially synthesized according to the sequence of the RBD on a cellulose membrane by using a MultiPep synthesizer (Intavis Bioanalytical Instruments, Germany) and SPOT technology (42, 43). The sequence of the RBD was obtained by GenBank (accession MN908947.3). Additional 15-mer peptides containing single mutations of variants of concern found within the RBD were designed. Single mutations included K417N, K417T, E484K, and N501Y (44). Detailed peptide sequences used in this study can be found in Table S2 in the supplemental material.

Synthesized arrays were probed with sera at a 1:400 dilution followed by incubation with horseradish peroxidase-conjugated goat anti-human IgA, IgG, and IgM polyclonal antibody at a 1:30,000 dilution. The bound of the secondary antibody on each peptide was detected and quantified by enhanced chemiluminescence. The peptide synthesis, probing, and quantification were outsourced to Kinexus Bioinformatics Corporation.

**Statistical analysis.** Chemiluminescence signal intensities of the peptide arrays were standardized in two ways, as follows: relative values to the maximum signal level of each array as 100, and Z-scores considering peptide signals of individual subjects as the population. These calculations were done by Microsoft Excel for Microsoft 365 MSO (16.0.14026.20202).

Nonlinear regression curve fitting was performed to calculate half-maximal neutralization titers (NT50s) of the neutralization assay. Statistical significance was calculated using unpaired two-tailed $t$ test. GraphPad Prism 9.1.0.221 was used for these statistical analyses.

The sequence and conformational information of the RBD was obtained under the accession no. 6M0J (5) and 7A94 (45) at Protein Data Bank (PDB). The images to depict the recognized epitopes are shown using The PyMOL (Molecular Graphics System, version 1.2r3pre; Schrödinger, LLC).

**Data availability.** The sequence used to design the peptide array was obtained under the accession number MN908947.3 at GenBank. The sequence of the peptides used in this study is available in the supplemental material of this article.

## SUPPLEMENTAL MATERIAL

Supplemental material is available online only.

**SUPPLEMENTAL FILE 1**, PDF file, 1.1 MB.

## ACKNOWLEDGMENTS

This work was funded by Japan Agency for Medical Research and Development (AMED) under grant no. JP20wm0125003 (Y.K.), JP20he1122001 (Y.K.), JP20nk0101627 (Y.K.), and JP20jk0110021 (Y. Nakagama). This work was also supported by JSPS KAKENHI grant no. JP21441824 (N.K.). We received support from Osaka City University's "Special Reserves" fund for COVID-19. We also received the COVID-19 Private Fund (to the Shinya Yamanaka laboratory, CiRA, Kyoto University). Y. Nitahara received the BIKEN Taniguchi Scholarship.

We thank healthy volunteers and patients who participated in this study.

We are grateful to National Institute of Infectious Diseases, Tokyo, Japan, for providing for the virus and to James A. Rankin for his contribution in checking the manuscript.

We declare no conflicts of interest.

Y. Nitahara, Y. Nakagama. N.K., and Y.K. designed the study. Y. Nitahara, Y. Nakagama, N.K., H.Y., Y. Mizobata, H.K., and Y.K. selected patients and acquired clinical data. Y. Nitahara, Y. Nakagama, N.K., K.C., Y. Michimuko, E.T.-K., and M.Y. performed immunological assays. Y. Nitahara, Y. Nakagama, N.K., and Y.K. performed epitope mapping analysis. Y. Nakagama and M.Y. performed neutralization assays. Y. Nitahara, Y. Nakagama, N.K., and Y.K. wrote the manuscript and contributed to analysis and interpretation of the data. H.Y., Y. Mizobata, H.K., A.K., and M.Y. contributed to critical discussion of the manuscript. All authors have read and approved the manuscript.

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
