## [Reviewer comments · Microbiology Spectrum]

Microbiology Spectrum

High resolution linear epitope mapping of the receptor binding domain of SARS-CoV-2 spike protein in COVID-19 mRNA vaccine recipients.

Yuko Nitahara, Yu Nakagama, Natsuko Kaku, Katherine Candray, Yu Michimuko, Evariste Tshibangu-Kabamba, Akira Kaneko, Hiromasa Yamamoto, Yasumitsu Mizobata, Hiroshi Takeya, Mayo Yasugi, and Yasutoshi Kido

Corresponding Author(s): Yasutoshi Kido, Osaka City University

Review Timeline:

Submission Date:	July 19, 2021
Editorial Decision:	August 24, 2021
Revision Received:	September 8, 2021
Accepted:	October 21, 2021

Editor: Daniel Perez

Reviewer(s): The reviewers have opted to remain anonymous.

Transaction Report:

DOI: <https://doi.org/10.1128/Spectrum.00965-21>

August 24, 2021

Dr. Yasutoshi Kido
Osaka City University
Osaka
Japan

Re: Spectrum00965-21 (High resolution linear epitope mapping of the receptor binding domain of SARS-CoV-2 spike protein in COVID-19 mRNA vaccine recipients.)

Dear Dr. Yasutoshi Kido:

Thank you for submitting your manuscript to Microbiology Spectrum. When submitting the revised version of your paper, please provide (1) point-by-point responses to the issues raised by the reviewers as file type "Response to Reviewers," not in your cover letter, and (2) a PDF file that indicates the changes from the original submission (by highlighting or underlining the changes) as file type "Marked Up Manuscript - For Review Only". Please use this link to submit your revised manuscript - we strongly recommend that you submit your paper within the next 60 days or reach out to me. Detailed information on submitting your revised paper are below.

Link Not Available

Sincerely,

Daniel Perez

Journals Department
Reviewer comments:

Reviewer #1 (Comments for the Author):

The authors compare a panel of linear and overlapping epitopes within the RBD of SARS-CoV-2 between vaccinated individuals and convalescent serum arising from natural infection. They find that epitope recognition is more generalized in vaccinated individuals, and more specific in previously infected patients. Consequently, they find that the quality of antibodies as measured by total RBD binding and viral neutralization is muted in vaccinated sera. The authors go on to perform epitope mapping on linear peptides matching variants of concern and find that the vaccinated sera outperform that from natural infections. Despite some awkward phrasing, I believe that the manuscript is well written and the work important.

"vaccine era" is awkward. Consider rephrasing.

Line 29 and 46: a vaccine is to a virus, not a disease. "...prompt rollout of the SARS-CoV-2 messenger RNA...)

Line 30: the use of "shall" is a bit formal and generally reserved for "I" or "We".which is becoming more dominant than natural infection mediated immunity.

Line 65. live or pseudotyped virus

Line 166: How were these individuals selected? Are they representative of all subjects within their own group? The color code for these individuals also needs to be added to figure 1's legend.

Line 174: Error in second residue numbering T415-F42?

Line 198: "two of them were sharing" is awkward. Consider two of them share the epitope...

Figure 2b: Antibody binding in naturally infected individuals is not visually compelling despite clear binding in figure 3b. Consider an upper limit.

Reviewer #2 (Comments for the Author):

The work by Nitahara et al (High resolution linear epitope mapping of the receptor binding domain of SARS-CoV-2 spike protein in COVID-19 mRNA vaccine recipients) examined the reactivity against a set of peptides representing different linear epitopes of the SARS-CoV-2 RBD. The authors used samples obtained from vaccinated people and COVID-19 patients (most of them critical patients) showing that COVID-19 patients possess higher neutralizing antibodies despite the similar levels of IgG antibodies between both groups. Then, using the peptide array a broader response against the RBD was detected in vaccinated group and higher reactivity was also detected in the presence of epitopes carrying some of the mutations described in variant of concern. Although the experiments are well designed and the information provided is valuable, there are substantial flaws about the interpretation of the results that require clarification. My specific comments are listed below:

1-My main concern about this work is that all the analysis regarding the COVID-19 patients were performed with 7 out of 10 samples obtained for critical patients. Although the authors did mention this limitation, the results could be heavily influenced by the patient condition. This is something that it could be easily evaluated since the authors possess a total of 20 samples from COVID-19 patients and samples from P08 to P20 are mild or moderate however, it was not performed. The data provided showed that in the case of P9 where a higher reactivity is observed along the different epitopes or the data regarding epitope 27, where high reactivity is observed in all patients with exception of P8, P9 and P10 (which are mild/moderate patients) reinforce the hypothesis that results could be influenced by the patient condition. Authors should evaluate the response from mild/moderate patients or redirect the manuscript towards vaccinated people vs critical COVID-19 patients comparison.

2-How an epitope was considered as recognized by just the vaccinated group, COVID-19 patients or both groups? The visualization analysis of figure 2B is not clear enough to understand the parameters used by the author to establish epitopes recognized by just one of the groups or both groups. In some cases, it is clear such as in the case of epitope 27 but overall, it is hard to agree with the authors in their classification. Looking more in detail the raw data provided (which is always appreciated), epitope 47 shows a reaction higher than 1,000 in just one COVID-19 patient which is like epitope 6. However, epitope 47 was classified as an epitope that reacts in both groups whereas epitope 6 was classified as an epitope recognized just by the vaccinated group. Overall, authors should provide a more detail explanation about the data analysis and how these classifications were established.

3-Why epitope 13 was not further analyzed despite the good reactivity observed?

4-A color scale for figure 2B like the one used in figure 6 will make the visualization of the results easier for the readers.

5-Please rephrase making this sentence shorter in the abstract ... 'Thus, relatively lower neutralizability was observed when a half-maximal neutralization titer measured in vitro by live virus neutralization assays was normalized to a total anti-RBD IgG titer'...

6-Figure 5C is not mentioned in the text

Staff Comments:

Preparing Revision Guidelines

To submit your modified manuscript, log onto the eJP submission site at <https://spectrum.msubmit.net/cgi-bin/main.plex>. Go to Author Tasks and click the appropriate manuscript title to begin the revision process. The information that you entered when you

first submitted the paper will be displayed. Please update the information as necessary. Here are a few examples of required updates that authors must address:

Please return the manuscript within 60 days; if you cannot complete the modification within this time period, please contact me. If you do not wish to modify the manuscript and prefer to submit it to another journal, please notify me of your decision immediately so that the manuscript may be formally withdrawn from consideration by Microbiology Spectrum.

If you would like to submit an image for consideration as the Featured Image for an issue, please contact Spectrum staff.

The authors compare a panel of linear and overlapping epitopes within the RBD of SARS-CoV-2 between vaccinated individuals and convalescent serum arising from natural infection. They find that epitope recognition is more generalized in vaccinated individuals, and more specific in previously infected patients. Consequently, they find that the quality of antibodies as measured by total RBD binding and viral neutralization is muted in vaccinated sera. The authors go on to perform epitope mapping on linear peptides matching variants of concern and find that the vaccinated sera outperform that from natural infections. Despite some awkward phrasing, I believe that the manuscript is well written, easy to follow, and the work important.

“vaccine era” is awkward. Consider rephrasing.

Line 29 and 46: a vaccine is to a virus, not a disease. “...prompt rollout of the SARS-CoV-2 messenger RNA...)

Line 30: the use of “shall” is a bit formal and generally reserved for “I” or “We”.which is becoming more dominant than natural infection mediated immunity.

Line 65. live or pseudotyped virus

Line 166: How were these individuals selected? Are they representative of all subjects within their own group? The color code for these individuals also needs to be added to figure 1’s legend.

Line 174: Error in second residue numbering T415–F42?

Line 198: “two of them were sharing” is awkward. Consider two of them share the epitope... Is there a reason figures 2,

Figure 2a: Antibody binding in naturally infected individuals is not visually compelling despite clear binding in figure 3b. Consider an upper limit.

Responses to the Reviewers' Comments

We appreciate your valuable advice for our manuscript. Here are our point-by-point responses to the reviewers' comments. All line numbers listed below are according to the ones in the marked-up version of the manuscript. All changes from the original manuscript are highlighted in red fonts.

Reviewer #1:

The authors compare a panel of linear and overlapping epitopes within the RBD of SARS-CoV-2 between vaccinated individuals and convalescent serum arising from natural infection. They find that epitope recognition is more generalized in vaccinated individuals, and more specific in previously infected patients. Consequently, they find that the quality of antibodies as measured by total RBD binding and viral neutralization is muted in vaccinated sera. The authors go on to perform epitope mapping on linear peptides matching variants of concern and find that the vaccinated sera outperform that from natural infections. Despite some awkward phrasing, I believe that the manuscript is well written and the work important.

1- "vaccine era" is awkward. Consider rephrasing.

Thank you, reviewer, for the thoughtful comment. We understand that the term "vaccine era" is not a widely accepted term, thus it is now modified as follows.

(Page 2, Line 31, Abstract)

"In the midst of coronavirus disease 2019 (COVID-19) vaccine deployment, understanding the epitope profiles of vaccine-elicited antibodies will be the first step in assessing functionality of vaccine-induced immunity."

(Page 3, Line 49, Importance)

"In the midst of vaccine deployment, it is of importance to describe the similarities and differences between the immune responses of COVID-19 vaccine recipients and naturally infected individuals."

(Page 3, Line 62, Introduction)

"In this start of acquiring vaccine immunity as a society against COVID-19, the *de novo* repertoire of vaccine-elicited antibodies in SARS-CoV-2 infection-naive individuals will be the first step to build an optimal host defense system towards vaccine-based population immunity."

2- Line 29 and 46: a vaccine is to a virus, not a disease. "...prompt rollout of the SARS-CoV-2 messenger RNA...)

3- Line 30: the use of "shall" is a bit formal and generally reserved for "I" or "We".which is becoming more dominant than natural infection mediated immunity.

4- Line 65. live or pseudotyped virus

We appreciate the thoughtful suggestions. The mentioned sentences are now modified as suggested.

(Page 2, Line 29, Abstract)

"The prompt rollout of the severe acute respiratory syndrome coronavirus 2 (SARS-CoV-2) messenger RNA (mRNA) vaccine is facilitating population immunity, which is becoming more dominant than natural infection mediated immunity."

(Page 3, Line 67, Introduction)

"Currently, the efficacy of vaccine-induced immunity against SARS-CoV-2 in an individual is evaluated by potential surrogate markers such as half-maximal neutralization titers (NT50) using live or pseudotyped viruses and total

antibodies titers against the receptor binding domain (RBD) of the spike protein of the virus.”

Together with the above modifications, the terms’ first usage in the Abstract was changed as follows.

(Page 2, Line 36, Abstract)

“In this study, the high-resolution linear epitope profiles of Pfizer-BioNTech COVID-19 mRNA vaccine recipients and COVID-19 patients were delineated by using microarrays mapped with overlapping peptides of the receptor binding domain (RBD) of ~~severe acute respiratory syndrome coronavirus 2~~ (SARS-CoV-2)-spike protein.”

5- Line 166: How were these individuals selected? Are they representative of all subjects within their own group? The color code for these individuals also needs to be added to figure 1's legend.

We thank the reviewer for the queries and the thoughtful suggestion. We clarified the selection of the analyzed individuals as follows.

(Page 8, Line 167, Results)

“To delineate the discrepancy in the epitope profiles between vaccine recipients and COVID-19 patients with high resolution, we next mapped and compared the immunodominant epitopes of ~~representative sera chosen from both groups~~ by using an overlapping 15-mer linear-peptide array (Figure 2a). ~~The representative sera were selected from vaccine recipients (N=5, median age 37 years) and COVID-19 patients (N=10, median age 58 years) based on their anti-RBD antibody titers so that the distribution of titers would represent the original population ($p=0.27$ and $p=0.12$, respectively. Supplement Table 1. Selected specimens are depicted as red dots in Figure 1).~~”

(Page 25, Line 568, Figure 1 Legend)

“~~Red dots denote the subset of samples chosen for the microarray analysis.~~”

Together, the below sentence was removed.

(Page 8, Line 173, Results)

“~~Sera of vaccine recipients and COVID-19 patients were incubated with the microarray. Sera of five subjects from the vaccine recipients and ten from the patients were selected based on their anti-RBD antibody titers and NT50 (denoted as red dots in Figure 1).~~”

6- Line 174: Error in second residue numbering T415-F42?

Thank you, reviewer, for pointing out the error. Now it is modified as follows with additional change in order of appearance.

(Page 9, Line 188, Results)

“Overall, seven linear epitopes were recognized within the RBD, four within (1): ~~V395-A411, peptide No.26-27; T415-F429, peptide No.33; V433-N450, peptide No.39-40; R457-S477, peptide No.47-49~~ and three within (2): N334-A348, peptide No.6; S373-L390, peptide No.19,20; S514-F541, peptide No.66-71, respectively (Figure 2b, Figure 3c).”

7- Line 198: "two of them were sharing" is awkward. Consider two of them share the epitope...

We are grateful for the reviewer’s thoughtful suggestion. The sentence is now modified as follows.

(Page 10, Line 213, Results)

“Three linear epitopes of the RBD were uniquely found in vaccine recipients’ sera (Figure 2b, Figure 3b, Figure 3c), two of them (peptide No.6 and No.19,20) shared the epitope regions of the RBD with neutralizing monoclonal antibodies known as class 3 and class 4.”

8- Figure 2b: Antibody binding in naturally infected individuals is not visually compelling despite clear binding in figure 3b. Consider an upper limit.

We appreciate your thoughtful advice on the visualization of the antibody binding in naturally infected individuals. In the Figure 2, the upper limit was set to 20,000 for both groups. We hope that this modification of the figure will make the binding peaks more compelling. The information was added to the Results and Figure 2b legend.

(Page 8, Line 181, Results)

“When calculating the relative signals, we set an upper limit of 20,000 relative light units for raw signals obtained in the assay for a better visualization in the heatmap. Four out of 1,065 peptides in total exceeded the limit (Supplement Table 3).”

(Page 25, Line 575, Figure 2b Legend)

“(b) Heat map identifying peptides recognized by IgG, IgA, and IgM in sera of vaccine recipients (Sample A–E) and COVID-19 patients (sample F–O). Signal of each peptide was normalized to the maximum signal in each subject. Raw signal data can be found in Supplement Table 3.

Reviewer #2:

The work by Nitahara et al (High resolution linear epitope mapping of the receptor binding domain of SARS-CoV-2 spike protein in COVID-19 mRNA vaccine recipients) examined the reactivity against a set of peptides representing different linear epitopes of the SARS-CoV-2 RBD. The authors used samples obtained from vaccinated people and COVID-19 patients (most of them critical patients) showing that COVID-19 patients possess higher neutralizing antibodies despite the similar levels of IgG antibodies between both groups. Then, using the peptide array a broader response against the RBD was detected in vaccinated group and higher reactivity was also detected in the presence of epitopes carrying some of the mutations described in variant of concern. Although the experiments are well designed and the information provided is valuable, there are substantial flaws about the interpretation of the results that require clarification. My specific comments are listed below:

1- My main concern about this work is that all the analysis regarding the COVID-19 patients were performed with 7 out of 10 samples obtained for critical patients. Although the authors did mention this limitation, the results could be heavily influenced by the patient condition. This is something that it could be easily evaluated since the authors possess a total of 20 samples from COVID-19 patients and samples from P08 to P20 are mild or moderate however, it was not performed. The data provided showed that in the case of P9 where a higher reactivity is observed along the different epitopes or the data regarding epitope 27, where high reactivity is observed in all patients with exception of P8, P9 and P10 (which are mild/moderate patients) reinforce the hypothesis that results could be influenced by the patient condition. Authors should evaluate the response from mild/moderate patients or redirect the manuscript towards vaccinated people vs critical COVID-19 patients comparison.

We appreciate the reviewer’s thoughtful suggestions and critical comments. As reviewer #1 has also raised a similar concern in his/her comment #5, we understand the reviewers’ concern as to the selection of the COVID-19 patients being skewed to critical condition of the disease. This limitation cannot be ignored in the analysis, nor is it our intention to mislead the readers. Therefore, we emphasized in the text multiple times to clarify the patients’ condition.

(Page 5, Line 92, Materials and Methods)

“Two groups were analyzed in this study: (i) vaccine recipients, all received two doses of BNT162b2 mRNA vaccine (Pfizer/BioNTech) with a three-week interval (N=21, age 20s–80s years old). Blood was obtained 17–28 days after the second dose (ii) **Mild to critical** COVID-19 patients confirmed by nucleic amplification testing (N=20, age 20s–80s years old). **We excluded asymptomatic patients in this study to observe the systematic immune response to the SARS-CoV-2. (19).**”

Together, new reference was added.

19. Takeshita M, Nishina N, Moriyama S, Takahashi Y, Uwamino Y, Nagata M, et al. Incomplete humoral response including neutralizing antibodies in asymptomatic to mild COVID-19 patients in Japan. *Virology*. 2021 Mar 1;555:35–43.

(Page 8, Line 167, Results)

“To delineate the discrepancy in the epitope profiles between vaccine recipients and COVID-19 patients with high resolution, we next mapped and compared the immunodominant epitopes **of representative sera chosen from both groups** by using an overlapping 15-mer linear-peptide array (Figure 2a). **The representative sera were selected from vaccine recipients (N=5, median 37 years old) and COVID-19 patients (N=10, median 58 years old) based on their anti-RBD antibody titers so that the distribution of titers would represent the original population ($p=0.27$ and $p=0.12$, respectively. Supplement Table 1. Selected specimens are depicted as red dots in Figure 1).**”

(Page 14, Line 305, Limitation)

“**The primary objective of our study was to compare the epitope profiles induced by vaccine and natural infection. However, the severity of the COVID-19 patients evaluated in the epitope analysis was skewed towards higher severity (one mild, two moderate and seven critical patients), whereas the vaccine recipients were relatively healthy without major comorbidities. Indeed, the epitope profiles of critical patients (P01–P07) seemed to differ from the ones of mild/moderate patients (P08–P10) (Figure 2b), suggesting the potential discrepancy in COVID-19 immune response depending on patients’ clinical conditions.**”

2-How an epitope was considered as recognized by just the vaccinated group, COVID-19 patients or both groups? The visualization analysis of figure 2B is not clear enough to understand the parameters used by the author to establish epitopes recognized by just one of the groups or both groups. In some cases, it is clear such as in the case of epitope 27 but overall, it is hard to agree with the authors in their classification. Looking more in detail the raw data provided (which is always appreciated), epitope 47 shows a reaction higher than 1,000 in just one COVID-19 patient which is like epitope 6. However, epitope 47 was classified as an epitope that reacts in both groups whereas epitope 6 was classified as an epitope recognized just by the vaccinated group. Overall, authors should provide a more detail explanation about the data analysis and how these classifications were established.

We are grateful for the reviewer’s critical comments and meaningful suggestions. We clarified the detection criteria in the text as below. Also, we additionally stated the substantial technical limitation when identifying epitopes in the microarray assay. References were added to supplement the readers’ understanding on membrane-based epitope mapping.

(Page 8, Line 185, Results)

“**We identified epitopes based on the following criteria: to have visibly detectable peaks when compared with adjacent peptides in the z-score depiction (Figure 3c) and to include peptide regions previously reported as epitopes of NAb in the literature (28, 29)**”

(Page 14, Line 313, Limitation)

~~“Moreover, the methodological limitation of the peptide binding assay using microarrays was that the results were semi-quantitative. Therefore, as mentioned the results, the reactive epitopes were characterized by combining local peak detection, not by a universal threshold, and steric characterizations (44, 45). In this manner, unreported and non-dominant potential epitopes were not recognized in this study.”~~

We added reference studies to which the same platform and methodology were applied for epitope detection.

44. Lima S de A, Guerra-Duarte C, Costal-Oliveira F, Mendes TM, Figueiredo LFM, Oliveira D, Avila RAM de, Ferrer VP, Trevisan-Silva D, Veiga SS, Minozzo JC, Kalapothakis E, Chávez-Olórtegui C. 2018. Recombinant Protein Containing B-Cell Epitopes of Different *Loxosceles* Spider Toxins Generates Neutralizing Antibodies in Immunized Rabbits. *Front Immunol* 9:3.

45. Klausberger M, Tscheliessnig R, Neff S, Nachbagauer R, Wohlbold TJ, Wilde M, Palmberger D, Krammer F, Jungbauer A, Grabherr R. 2016. Globular Head-Displayed Conserved Influenza H1 Hemagglutinin Stalk Epitopes Confer Protection against Heterologous H1N1 Virus. *PLoS One* 11:e0153579.

Together, the below sentences were removed.

(Page 6, Line 134, Materials and Methods)

~~“The epitopes were detected by subjective visual inspection. Our cutoff for signal detection was set at visually detectable peaks in a graph depicting a mostly minimum of 0.5 z-score of the mean peptide signals and/or regions previously reported as neutralizing antibodies in the RBD (26,27).”~~

3-Why epitope 13 was not further analyzed despite the good reactivity observed?

Thank you, reviewer, for the meaningful comment. We recognized the mentioned epitope and had considered that it would only partially match the previously reported neutralizing monoclonal antibody S309. Below was added to clarify the analysis.

(Page 10, Line 219, Results)

~~“At N-terminus of the RBD, namely the peptide No.1–6, we identified an epitope region detected only in the post-vaccination sera (Figure 2b, Figure 3c). Especially, peptide No. 6 was an epitope of note, which ~~could shall~~ be recognized by the class3 NAb S309 by P337–A344 helix residues (Figure 5a) (32). The epitope is distinct from the receptor-binding motif and has a good accessibility both in the up and down compositions of the RBD (PDB, 7A49, Figure 5b). **Reactive peptide No.13 was located close to the peptide No.6 in the steric conformation (Figure 2b, Figure 3c). The peptide No.13 contained six residues (K356–C361), which share conformational epitope of the NAb S309 with the peptide No.6 (28,32). Nevertheless, the peptide No.13 alone does not seem to make a dominant epitope of S309.”**~~

4-A color scale for figure 2B like the one used in figure 6 will make the visualization of the results easier for the readers.

Thank you, reviewer, for the suggestion. We changed to use the same color scale for the figure 2b as shown below.

5-Please rephrase making this sentence shorter in the abstract ... ‘Thus, relatively lower neutralizability was observed when a half-maximal neutralization titer measured *in vitro* by live virus neutralization assays was normalized to a total anti-RBD IgG titer’...

We appreciate the reviewer for the meaningful comment. Now we split the sentence to two parts as follows. We hope this change will make the understanding easier for the readers.

(Page 2, Line 38, Abstract)

“Half-maximal neutralization titers were measured *in vitro* by live virus neutralization assays. As a result, relatively lower neutralizability was observed in vaccine recipients’ sera when it was normalized to the total anti-RBD IgG titer.”

6-Figure 5C is not mentioned in the text

Thank you, reviewer, for the thoughtful comment. Now it is mentioned in the text as follows.

(Page 10, Line 224, Results)

“Another identified epitope, peptide No.19 and 20 (Figure 5c), shared epitope residues with a neutralizing monoclonal antibody CR3022, categorized as class 4, isolated from a SARS-CoV convalescent (32,33).”

(Page 12, Line 260, Discussion)

“In our study, two classes of NAb exclusively relevant in vaccine-elicited sera were found to be of specific note; peptide No.6 (Figure 5a,b) targeted by the class 3 NAb (31) and peptide No.19–20 (Figure 5c) targeted by the class 4 NAb (33,34).”

(Page 12, Line 267, Discussion)

“To the contrary, the linear peptides No.6 and No.19–20 (Figure 5a, Figure 5b, Figure 5c), corresponding to epitopes found exclusively in vaccine-elicited sera (Figure 2b, Figure 3c), revealed corresponding epitopes targeted by human NAb isolated from SAR-CoV convalescent (S309 and CR3022, respectively) (31,33,34).”

October 21, 2021

Dr. Yasutoshi Kido
Osaka City University
Osaka
Japan

Re: Spectrum00965-21R1 (High resolution linear epitope mapping of the receptor binding domain of SARS-CoV-2 spike protein in COVID-19 mRNA vaccine recipients.)

Dear Dr. Yasutoshi Kido:

Your manuscript has been accepted, and I am forwarding it to the ASM Journals Department for publication. You will be notified when your proofs are ready to be viewed.

Sincerely,

Daniel Perez
Editor, Microbiology Spectrum

Journals Department
Supplemental Material FOR Publication: Accept